# Continual Learning for Multilingual Neural Machine Translation via Dual Importance-based Model Division

**Junpeng Liu[1]**     **Kaiyu Huang[2]**     **Hao Yu[1]**
**Jiuyi Li[1]**     **Jinsong Su[3]**     **Degen Huang[1]**[*]

[1]Dalian University of Technology
[2]Institute for AI Industry Research, Tsinghua University     [3]Xiamen University
{liujunpeng_nlp, yuhao_dlut, lee.91}@mail.dlut.edu.cn
huangkaiyu@air.tsinghua.edu.cn
jssu@xmu.edu.cn     huangdg@dlut.edu.cn

## Abstract

A persistent goal of multilingual neural machine translation (MNMT) is to continually adapt the model to support new language pairs or improve some current language pairs without accessing the previous training data. To achieve this, the existing methods primarily focus on preventing catastrophic forgetting by making compromises between the original and new language pairs, leading to sub-optimal performance on both translation tasks. To mitigate this problem, we propose a dual importance-based model division method to divide the model parameters into two parts and separately model the translation of the original and new tasks. Specifically, we first remove the parameters that are negligible to the original tasks but essential to the new tasks to obtain a pruned model, which is responsible for the original translation tasks. Then we expand the pruned model with external parameters and fine-tune the newly added parameters with new training data. The whole fine-tuned model will be used for the new translation tasks. Experimental results show that our method can efficiently adapt the original model to various new translation tasks while retaining the performance of the original tasks. Further analyses demonstrate that our method consistently outperforms several strong baselines under different incremental translation scenarios. [1]

## 1 Introduction

Multilingual neural machine translation (MNMT) (Johnson et al., 2017; Zhang et al., 2020; Liu et al., 2022; Goyal et al., 2022) handles multiple translation directions in a single model and recent large-scale MNMT models, such as mBART50-nn (Tang et al., 2020), M2M-100 (Fan et al., 2021) and NLLB (Costa-jussà et al., 2022) have demonstrated promising translation performance. However, a

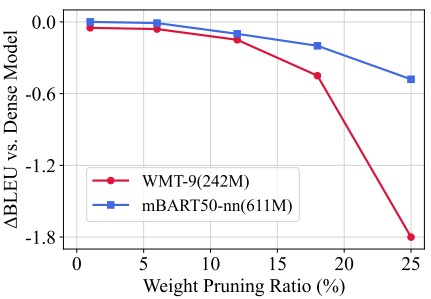

Figure 1: The BLEU decline of different MNMT models on the *en↔xx* under different pruning ratios. We perform weight pruning with the general magnitude-based method (See et al., 2016).

practical problem is how to efficiently extend those MNMT models to support new languages or improve some existing translation directions when the new training data is available. Considering that the current MNMT models are generally trained with a large amount of data (Liu et al., 2021; Ebrahimi and Kann, 2021), it will be extremely resource and time-consuming to retrain the model from scratch. Therefore, the method that can continually adapt the previously learned model to new translation tasks while retaining the original knowledge without accessing the previous training data, i.e., continual learning, has drawn increasing research attention over the past few years (Gu et al., 2022; Zhao et al., 2022; Sun et al., 2023; Huang et al., 2023).

The biggest challenge for continual learning is catastrophic forgetting (French, 1993), which results in severe performance decline on the previous tasks when adapting to new tasks. To alleviate this problem, some works attempt to balance the performance between the original and incremental tasks, such as the replay-based methods (Ko et al., 2021; Liu et al., 2021; Garcia et al., 2021) and regularization methods (Khayrallah et al., 2018; Thompson et al., 2019; Zhao et al., 2022). However, as the whole parameter spaces are shared across different

---

[*]Corresponding Author
[1]Our code is available at https://github.com/raburabu91/BVP4CL

translation tasks, those methods are vulnerable to inter-task interference, especially when the incremental tasks are largely different from the original ones. Other works (Bapna and Firat, 2019; Escolano et al., 2021) employ additional task-specific parameters for new task adaptation while keeping the parameters in the original model fixed, which can naturally avoid the forgetting problem. However, those methods increase the model size and the structure of the task-specific modules requires specialized manual design.

Previous studies (Bau et al., 2019; Voita et al., 2019; Gu et al., 2021; Liang et al., 2021) show that some parameters in the network are not important to the original tasks so that they can be pruned without causing obvious performance degradation. As shown in Figure 1, the current MNMT models also have such unimportant parameters, which provides the possibility of using those negligible parameters to learn the knowledge of new tasks instead of introducing additional parameters. However, this model pruning strategy focuses more on the importance of parameters to the original tasks while neglecting their roles in new task adaptation, which restricts the model's ability to learn the specific knowledge of the incremental languages.

In this work, considering the importance of parameters on the original and new translation tasks, we propose a dual importance-based model division method that can efficiently adapt the original MNMT model to various incremental language pairs. First, we fine-tune the original MNMT model with the new training data and propose two methods to evaluate the importance of parameters in different tasks. Then we find the parameters that are unimportant to the original tasks but important to the new tasks and remove them from the original model to obtain a pruned model. Finally, we expand the pruned model to its original size and fine-tune the newly added parameters with new training data. In this way, we can better capture the specific knowledge of incremental translation tasks and avoid catastrophic forgetting on the original translation tasks. Our main contributions are summarized as follows:

- We propose a dual importance-based model division method to improve the model's ability to learn the knowledge of new language pairs, which achieves competitive translation qualities on incremental translation tasks.

- Our method can efficiently reorganize the roles of the model parameters in incremental learning and preserve the performance of the previous translation tasks without accessing the original training data.

- We conduct extensive experiments in different incremental translation scenarios. Experimental results show that our method can be easily applied to the pre-trained MNMT models and improve the zero-shot translation.

## 2 Related Work

**Replay-based Methods.** The replay-based methods utilize the training data or pseudo data of the previous tasks to replay the old knowledge while training on new tasks (Lakew et al., 2018; Sun et al., 2019; de Masson D'Autume et al., 2019; Liu et al., 2021; Garcia et al., 2021). However, those methods will bring more training consumption, especially for the large-scale pre-trained MNMT model. Moreover, the previous training data are sometimes unavailable due to data privacy or storage limitations (Feyisetan et al., 2020; Gu et al., 2022). The noise in pseudo data will also hurt the performance of the previous and new tasks. Compared to those methods, our method does not need to access the previous data, and thus it is more flexible and efficient for incremental learning.

**Regularization-based Methods.** In addition to the original training objective, the regularization-based methods usually introduce additional penalty terms to balance the performance on the previous and new tasks. Khayrallah et al. (2018) and Thompson et al. (2019) employ regularization terms to constrain the change of all the parameters from their original values. Castellucci et al. (2021) utilizes the original model as the teacher to prevent catastrophic forgetting via knowledge distillation. Gu et al. (2022) updates the parameters within the low forgetting risk regions with a hard constraint. By contrast, our method only prunes the parameters that are not essential to the previous tasks so that it can prevent the forgetting problem. It also allows our method to focus on learning the new tasks without making compromises with the previous tasks.

**Parameter Isolation-based Methods.** The parameter isolation-based methods generally allocate the model parameters to different tasks. Of this kind, some works (Bapna and Firat, 2019; Madotto

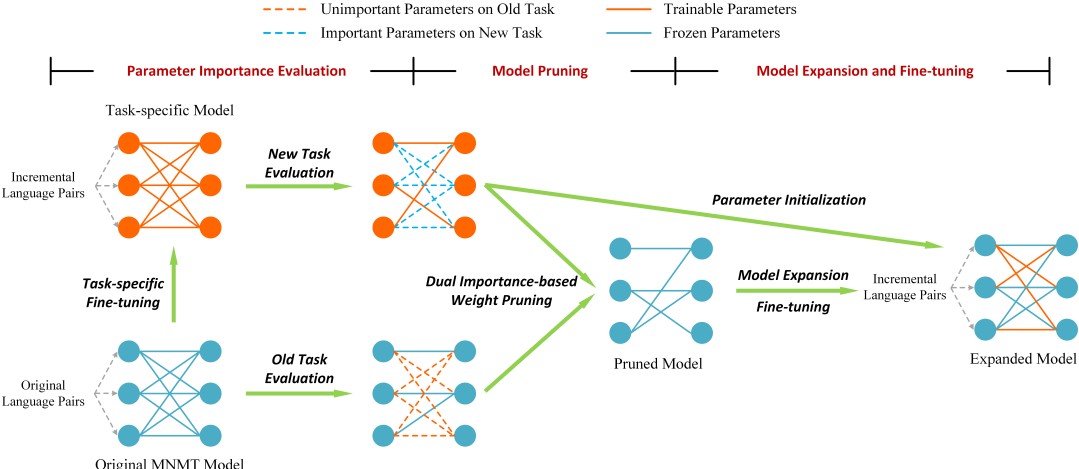

Figure 2: The training process of the proposed method. Our method consists of three steps: parameter importance evaluation, model pruning and model expansion & fine-tuning.

et al., 2021; Zhu et al., 2021) add extra task-specific parameters to the original model for new task learning. Despite retaining the performance on previous tasks, those methods inevitably increase the number of model parameters. Other works (Liang et al., 2021; Gu et al., 2021) prune the model based on the parameter importance to the original tasks for domain adaptation. Different from those methods, our method explores the potentiality of task-specific parameters to improve the performance and efficiency of the new task adaptation without additional parameters.

**Model Pruning.** Model Pruning usually aims to reduce the model size and has been widely used in natural language processing tasks. Sun et al. (2020) highlights the importance of the sparse network structure. Lin et al. (2021) focuses on finding the important parameters for each language to perform language-specific modeling in MNMT. Gu et al. (2021) utilizes the unimportant parameters to learn the in-domain knowledge. Compared to those methods, our method considers the importance of parameters on the previous and incremental tasks simultaneously during the model pruning stage, which yields better performance on new tasks while retraining the knowledge of the previous tasks.

## 3 Method

In this work, we aim to prevent the catastrophic performance degradation on the previous translation tasks without accessing their training data while adapting the model to new translation tasks efficiently. The main idea of our method is that the

parameters have varying degrees of importance for the previous and new translation tasks. Based on this, we utilize the parameters that are not important to the previous tasks but essential to the new tasks to perform new task adaptation. Specifically, our method involves the following steps as shown in Figure 2. First, we fine-tune the original MNMT model on the new training data to obtain a task-specific model, and then evaluate the importance of parameters in the original MNMT model and the task-specific model separately. Second, we remove the parameters that are unimportant to the original tasks but important to the new tasks from the original MNMT model to obtain a pruned model, and we make it responsible for the translation of the original tasks. The pruned model can retain the performance of the original tasks since the removed parameters are not necessary. Last, we expand the pruned model to its original size and fine-tune the newly added parameters to learn the new translation tasks. As those new parameters are essential to the new tasks, they can learn the knowledge of the new tasks more efficiently.

### 3.1 Importance Evaluation

The importance evaluation procedure is to search for the negligible parameters in the original model that can be erased and the essential parameters that contribute more to the new task adaptation. To achieve this, we perform importance evaluation for the original and new translation tasks separately.

**Original Task Evaluation.** To evaluate the parameter importance in the original model, we adopt the general magnitude-based approach (See et al.,

2016) which regards the absolute value of each parameter as its own importance:

$$I_v(w^o_{ij}) = |w^o_{ij}|, w^o_{ij} \in \mathbf{W^o} \qquad (1)$$

where $w^o_{ij}$ denotes the $i$-th row and $j$-th column parameter of a certain weight matrix $\mathbf{W^o}$ in the original model. In this way, the smaller the absolute value, the less important the parameter is.

**New Task Evaluation.** To obtain the important parameters for the new translation tasks, we first fine-tune the original model with the new training data to obtain a task-specific model. Considering that the new languages may have different scripts from the original languages, we extend the embedding layer of the original model to avoid the out-of-vocabulary problem. The vocabulary extension method is depicted in Appendix B. After that, we propose two evaluation methods based on the magnitude and variation of the parameters, respectively. On the one hand, fine-tuning the original model on the new translation tasks will amplify the magnitude of the important parameters and diminish that of the unimportant parameters. Therefore, we can simply utilize a similar magnitude-based method to Equation (1) for the new task evaluation. On the other hand, the variation of each parameter from the original MNMT model to the task-specific model can also illustrate their importance. Intuitively, the value variations of the important parameters should be larger than those of the unimportant parameters since they capture the specific knowledge of the new tasks. Based on this, the variation-based evaluation method is formulated as

$$I_c(w^s_{ij}) = |w^s_{ij} - w^o_{ij}|, w^s_{ij} \in \mathbf{W^s}, w^o_{ij} \in \mathbf{W^o} \quad (2)$$

where $w^o_{ij}$ and $w^s_{ij}$ represent the $i$-th row and $j$-th column parameter of the weight matrices $\mathbf{W^o}$ and $\mathbf{W^s}$. The weight matrices $\mathbf{W^o}$ and $\mathbf{W^s}$ denote different parts of the original and fine-tuned specific model, respectively.

## 3.2 Dual Importance-based Model Pruning

Based on the importance of parameters to the original and new translation tasks, we determine which parameters in the original MNMT model should be pruned. To make the most of the model's ability to learn the knowledge of new tasks while preserving the performance on the original tasks, we propose a dual importance-based model pruning method.

Specifically, based on the importance evaluation results on the original tasks, we first select $a\%$ parameters with the smallest magnitude from each weight matrix $\mathbf{W^o}$ to form the pruning candidate set $\mathcal{M}$. Since the parameters in $\mathcal{M}$ are relatively unimportant to the original tasks, they can be pruned without causing catastrophic forgetting. Then we determine which parameters in $\mathcal{M}$ should be finally pruned based on the importance evaluation results on the new tasks. According to different importance evaluation methods for the new tasks and pruning distributions in each weight matrix, we propose three model pruning strategies.

**Uniform Magnitude-based Pruning (UMP)** adopts the magnitude-based method and sorts the parameters in $\mathcal{M}$ by their magnitude in weight matrix $\mathbf{W^s}$, then $\rho\%$ parameters with largest magnitude in each weight matrix will be pruned ($\rho < a$).

**Uniform Variation-based Pruning (UVP)** adopts a similar pruning method to UMP except that the importance evaluation method is changed to the variation-based one.

**Blind Variation-based Pruning (BVP)** also adopts the variation-based method, but different from UVP, it selects $b\%$ parameters with the largest variation from each weight matrix $\mathbf{W^s}$ to form the important parameter set $\mathcal{N}$. Then the parameters in the intersection set of $\mathcal{M}$ and $\mathcal{N}$ will be pruned.

Note that the pruning ratios in each weight matrix are the same for the first two methods while varying for the third method. The pruned model will be used to generate the translation of the original language pairs in the inference stage.

## 3.3 Transfer-based Model Expansion

In this step, we add new parameters to the pruned model and expand it to the original size. Moreover, the embedding layer of the pruned model is also extended with the embeddings of new languages similar to the task-specific fine-tuning in Section 3.1. Considering the potential representation gap between the newly added parameters/embeddings and the parameters in the pruned model, we propose a transfer-based parameter adaptation that initializes the new parameters/embeddings with their counterparts in the fine-tuned task-specific model. On the one hand, those parameters have been fitted well with the original model during the task-specific fine-tuning stage so that they do not suffer from the representation gap. On the other hand, the task-specific knowledge contained in those parameters

will benefit the new task adaptation. We fine-tune the expanded model on the new training data ($\mathcal{D}$) and the training objective is

$$\mathcal{L}(\theta_O, \theta_N) = \sum_{(\mathbf{x},\mathbf{y}) \in \mathcal{D}} \log p(\mathbf{y}|\mathbf{x}; \theta_O, \theta_N) \quad (3)$$

where $\theta_N$ denotes the trainable parameters including the newly added parameters ($\theta_P$) and embeddings ($\theta_E$), and $\theta_O$ denotes the pruned model which is kept frozen during this training stage. In the inference stage, all the parameters in the expanded model are employed to generate the translation of the new tasks.

## 4 Experiments

### 4.1 Datasets

To ensure the reliability of the experiments, we first perform continual learning based on a self-trained MNMT model and then employ the pre-trained mBART50-nn model (Tang et al., 2020) as the initial model to evaluate our method in the real-world scenario. For the former, the original MNMT model is trained on a multilingual translation dataset (WMT-9) covering 8 language pairs. And then we select another four language pairs for incremental learning. For the latter, we adopt mBART50-nn as the initial MNMT model and perform continual learning in two translation tasks: the language adaptation task and the language enhancement task. The language adaptation task aims to enable the model to support the translation of new languages, while the language enhancement task aims to improve the translation performance of some existing language pairs that are already supported by the model. The detailed descriptions of datasets for the original and incremental languages are in Appendix A.

### 4.2 Implementation Details

**Baselines.** The original MNMT model is trained based on the vanilla Transformer (Vaswani et al., 2017) architecture with multiple parallel data following Johnson et al. (2017). We compare our method with various previous methods in continual learning. The baselines are as follows:

- **Scratch** This method trains the initial MNMT model on the original 8 language pairs for continual learning and the bilingual model on each incremental language pair from scratch.

- **Replay** (Sun et al., 2019) This method creates pseudo training data for the original language pairs and trains new models on the combination of the pseudo training data and incremental training data.

- **Fine-Tuning** (Luong and Manning, 2015) This model is trained based on the original MNMT model only with the incremental training data.

- **L2-Reg** (Miceli Barone et al., 2017) This method adds an additional L2-norm regularization term to the training objective.

- **EWC** (Kirkpatrick et al., 2017) This method evaluates the importance of parameters with the Fisher information matrix and adds a penalty term to the training objective to retain the original knowledge.

- **LFR** (Gu et al., 2022) This method updates the parameters in the low forgetting risk regions with a hard constraint. We reproduce the LFR-OM model in the experiments.

- **Adapter** (Bapna and Firat, 2019) This method freezes the whole network of the original model and employs additional networks to learn the knowledge of incremental language pairs. The dimension of the projection layer is set to 128 in the experiments.

- **PTE** (Gu et al., 2021) This method compresses the original model to learn new tasks and introduces a distillation loss to retain the old knowledge. We omit the knowledge distillation step as the original training data is not available in our settings.

Other baselines in the experiments based on the mBART50-nn model are as follows:

- **mBART50-nn** (Tang et al., 2020) The large-scale pre-trained MNMT model, which is the baseline model for the language enhancement task. Other systems are implemented based on this model.

- **mBART50-nn+LSE** (Berard, 2021) This method inserts a new language-specific embedding layer (LSE) for the new languages. we adopt it as the baseline model for the language adaptation task and implement other methods based on it.

| Model | Original Language Pairs | | | Incremental Language Pairs | | | | | | | | |
|---|---|---|---|---|---|---|---|---|---|---|---|---|
| | en→xx | xx→en | AVG1 | en→ro | ro→en | en→de | de→en | en→ta | ta→en | en→ga | ga→en | AVG2 |
| Scratch | 18.90 | 23.21 | 21.06 | 23.42 | 31.29 | 24.78 | 30.13 | 9.95 | 14.41 | 12.43 | 12.61 | 19.88 |
| Fine-Tuning | 0.50 | 0.50 | 0.50 | 24.56 | 33.81 | 25.39 | 30.61 | 10.43 | 15.63 | 19.09 | 21.01 | 22.57 |
| Replay | 15.27 | 22.90 | 19.09 | 19.75 | 29.66 | 22.16 | 28.45 | 10.14 | 15.10 | 17.06 | 19.88 | 20.28 |
| L2-Reg | 14.80 | 23.89 | 19.35 | 21.94 | 31.83 | 17.87 | 27.27 | 8.53 | 16.40 | 12.97 | 20.69 | 19.69 |
| EWC | 16.40 | 23.32 | 19.86 | 22.77 | 32.18 | 19.12 | 27.52 | 8.68 | 16.84 | 13.40 | 21.41 | 20.24 |
| LFR | 17.35 | 23.43 | 20.39 | 22.90 | 32.46 | 18.93 | 27.91 | 9.12 | **16.98** | 13.54 | 21.55 | 20.42 |
| Adapter | **18.90** | **23.21** | **21.06** | 23.89 | 33.36 | 21.89 | 29.14 | 9.49 | 16.31 | 16.01 | 21.81 | 21.49 |
| PTE | 18.87 | 23.04 | 21.01 | 24.03 | 33.42 | 22.09 | 28.65 | 9.69 | 16.46 | 17.55 | 21.46 | 21.67 |
| UMP | 18.69 | 22.63 | 20.66 | **24.70** | 33.78 | 22.64 | **29.31** | 9.54 | 16.19 | 18.53 | **22.89** | 22.19 |
| UVP | 18.85 | 23.09 | 20.97 | 24.63 | 33.82 | 22.76 | 29.19 | 9.68 | 16.73 | **19.58** | 22.37 | 22.35 |
| BVP | 18.86 | 23.10 | 20.98 | 24.60 | **33.98** | **22.94** | 29.13 | **9.92** | 16.69 | 19.36 | 22.42 | **22.38** |

Table 1: The BLEU scores of adding a single language pair for MNMT in incremental learning. "AVG1" and "AVG2" denote the average BLEU scores on the original language pairs and incremental language pairs, respectively. The best BLEU scores among all the incremental learning methods are marked in bold.

**Training Setup.** Our translation models are built on the Transformer using the open-source Fairseq implementation (Ott et al., 2019)[2]. We employ the same configuration of Transformer-Big (Vaswani et al., 2017) in our experiments for fair comparisons. The details about model settings are in Appendix B.

**Evaluation.** We report the detokenized case-sensitive BLEU offered by SacreBLEU (Post, 2018)[3]. We employ the best checkpoint on the validation dataset for evaluation, and perform beam search decoding with a beam size of 4 and length penalty of 1.0.

### 4.3 Main Results

**Adapting to a single language pair.** We first study the translation performance when adapting the original MNMT to a single language pair. The results are summarized in Table 1. Compared with the Scratch model, our method achieves remarkable performance improvement on the incremental language pairs (up to +2.50 BLEU on average) and the translation qualities are also competitive with the Fine-Tuning model. Moreover, our method prevents catastrophic forgetting on the original language pairs. Despite prior attempts by the replay-based and regularization-based methods to make a balance between the original and incremental language pairs, they still suffer from obvious performance degradation on the original en→xx translation tasks. Although no pronounced degradation has occurred in the Adapter and PTE models, they

are weak in their ability to learn the knowledge of new language pairs, leading to a performance gap compared to the Fine-Tuning model. In contrast to those prior continual learning methods, our method achieves the best overall translation performance.

**Adapting to multiple language pairs.** We next investigate the translation qualities when adapting the original MNMT model to multiple language pairs simultaneously. The results are summarized in Table 2. In this scenario, the regularization-based methods retain their performance on the original language pairs at the cost of the insufficient ability to learn the incremental language pairs, which leads to inferior performance to the Scratch model. We also find that the performance degradation on English-to-Irish (en→ga) translation is more severe than other language pairs. We ascribe this to the larger linguistic difference between Irish (ga) and other languages, as Irish has a distinct dominant word order (VSO) from other languages (SVO or SOV). This difference may lead to interference across languages, which increases the difficulty of the regularization-based methods to balance the translation performance between the original and incremental language pairs. By contrast, our method can better deal with the increased language divergence, and yield superior translation performance to the Scratch and PTE models. As we consider the incremental language pairs during the parameter importance evaluation stage, our method can capture the specific knowledge of each incremental language pair more efficiently than other methods.

---
[2] https://github.com/pytorch/fairseq
[3] Signature: BLEU+case.mixed+numrefs.1+smooth.exp+tok.13a+version.1.5.1.

| Model | Original Language Pairs | | | Incremental Language Pairs | | | | | | | | |
|---|---|---|---|---|---|---|---|---|---|---|---|---|
| | en→xx | xx→en | AVG1 | en→ro | ro→en | en→de | de→en | en→ta | ta→en | en→ga | ga→en | AVG2 |
| Scratch | 18.90 | 23.21 | 21.06 | 23.42 | 31.29 | 24.78 | 30.13 | 9.95 | 14.41 | 12.43 | 12.61 | 19.88 |
| Fine-Tuning | 0.40 | 1.74 | 1.07 | 24.68 | 34.85 | 26.08 | 31.34 | 10.11 | 16.81 | 14.03 | 22.81 | 22.59 |
| L2-Reg | 16.82 | **23.60** | 20.21 | 16.55 | 25.51 | 14.66 | 25.44 | 6.26 | 14.08 | 2.63 | 9.88 | 14.38 |
| EWC | 16.97 | 23.46 | 20.22 | 17.00 | 26.54 | 18.50 | 26.99 | 6.50 | 14.08 | 2.51 | 9.42 | 15.19 |
| LFR | 17.16 | 23.49 | 20.33 | 18.21 | 28.58 | 16.56 | 26.24 | 6.56 | 14.37 | 4.20 | 12.75 | 15.93 |
| PTE | 18.79 | 22.80 | 20.79 | 23.06 | 32.16 | 20.47 | 27.79 | 8.12 | 15.38 | 12.04 | 20.12 | 19.89 |
| UMP | 18.74 | 22.85 | 20.80 | 23.27 | **33.09** | 21.08 | 28.49 | 8.84 | **16.02** | 12.40 | 21.20 | 20.55 |
| UVP | **18.87** | 23.11 | 20.99 | **23.54** | 32.92 | 21.23 | **28.61** | **9.05** | 15.23 | 12.74 | 20.94 | 20.53 |
| BVP | 18.85 | 23.16 | **21.01** | 23.53 | 32.92 | **21.34** | 28.49 | 8.97 | 15.88 | **12.78** | **21.39** | **20.66** |

Table 2: The BLEU scores of adding all eight language pairs simultaneously for MNMT in incremental learning. The best results among all the continual learning methods are highlighted in bold.

## 5 Analysis

### 5.1 Ablation Studies

**Effects of Different Pruning Ratios.** To better balance the translation performance on the original and incremental translation tasks, we quantify the trade-off between the pruning ratio and the translation performance. The results are depicted in Figure 3. We also plot the PTE model in the figure for comprehensive comparisons. For each method, the performance on the original language pairs degrades with the increase of the pruning ratio, while the results on the incremental language pairs are quite reversed. Specifically, the proposed three pruning methods consistently outperform the PTE model on the incremental language pairs under the same pruning ratios. Moreover, the UVP and BVP models can achieve better results with a smaller pruning ratio of 6% than the PTE model of 18%. These results demonstrate that training the incremental language pairs with their corresponding important parameters can improve the efficiency of continual learning, which makes it possible to adapt the original model to more new translation tasks. The UVP and BVP models can better retain the performance on the original language pairs while the UMP model suffers from more apparent performance degradation. Considering the overall performance under different settings, we employ the BVP model for the following experiments.

**Effects of Dual importance-based Pruning.** We study the dual-importance pruning strategy with our BVP method under the pruning ratio of 6%. As shown in Table 3, when pruning only based on the old task importance, the model can maintain its performance on the original language pairs. However, it is weak in the ability of capturing the specific

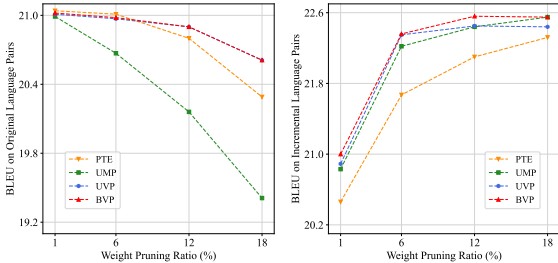

(a) Original Languages  (b) Incremental Languages

Figure 3: Comparisons of PTE, UMP, UVP and BVP under different pruning ratios.

| Method | en→xx | xx→en | en↔ro | en↔de | en↔ta | en↔ga |
|---|---|---|---|---|---|---|
| Old | 18.87 | 23.04 | 28.73 | 25.37 | 13.08 | 19.51 |
| New | 0.03 | 0.04 | 29.36 | 26.98 | 13.07 | 20.96 |
| BVP | 18.86 | 23.10 | 29.29 | 26.04 | 13.31 | 20.89 |

Table 3: Ablation study on dual importance-based pruning. "Old" and "New" indicate the model pruning methods are only based on the parameter importance on the old task and new task, respectively.

knowledge of new language pairs, leading to inferior performance to our BVP model. In contrast, when pruning only based on the new task importance, although the model performs slightly better than BVP on some new language pairs, it fails on the original language pairs. We find that most (>98%) pruned parameters in this method are also important to the old language pairs. Therefore, the performance on original language pairs decreases apparently if those parameters are pruned. Our BVP model achieves the best overall performance across all language pairs, showing the necessity of the dual-importance.

**Effects of Different Initialization methods.** We implement three initialization methods to investi-

| Model | $en{\leftrightarrow}xx$ | $en{\rightarrow}yy$ | $xx{\rightarrow}yy$ |
|---|---|---|---|
| Scratch | 21.06 | 17.65 | – |
| Fine-Tuning | 2.25 | 19.51 | 5.18 |
| L2-Reg | 17.27 | 14.46 | 3.54 |
| EWC | 18.45 | 15.93 | 3.67 |
| LFR | 18.72 | 16.69 | 3.84 |
| PTE | **20.98** | 18.12 | 6.16 |
| Ours(BVP) | **20.98** | **18.85** | **6.67** |

Table 4: The BLEU results on zero-shot translation. "*xx*" represent the 8 original languages and "*yy*" represent the 4 incremental languages in Table 9. The best results among all the continual learning methods are in bold.

| Model | $en{\leftrightarrow}xx$ | $en{\leftrightarrow}ro$ | $en{\leftrightarrow}ta$ |
|---|---|---|---|
| Scratch | $21.06_{(-0.00)}$ | $27.36_{(-0.00)}$ | 12.18 |
| Fine-Tuning | $0.31_{(-0.19)}$ | $0.34_{(-28.85)}$ | 12.86 |
| L2-Reg | $19.17_{(-0.18)}$ | $26.52_{(-0.37)}$ | 11.90 |
| EWC | $19.39_{(-0.47)}$ | $26.59_{(-0.89)}$ | 12.17 |
| LFR | $19.37_{(-1.02)}$ | $26.81_{(-0.87)}$ | 12.27 |
| PTE | $20.89_{(-0.17)}$ | $28.42_{(-0.31)}$ | 12.59 |
| Ours(BVP) | $\mathbf{20.90}_{(-0.08)}$ | $\mathbf{29.18}_{(-0.11)}$ | **13.05** |

Table 5: The BLEU results of the sequential language incremental learning. The value in brackets represents the performance decline before and after training with the $en{\leftrightarrow}ta$ data. The best results among all the continual learning methods are in bold.

gate the effects of different parameter initialization methods in the model expansion stage. The results indicate that initializing the newly added parameters with their counterparts in the task-specific fine-tuned model enables the BVP model to attain optimal translation performance with minimal training time for incremental learning. The detailed comparisons and analyses are provided in Appendix C.

## 5.2 Results on Zero-shot Translation

To investigate the transfer ability of our method, we further conduct zero-shot translation from the original languages to the incremental languages. To make the most of the ability to learn the knowledge of the incremental languages, we adapt the original MNMT model to only one incremental language each time. We adopt the Flores (Goyal et al., 2022) test sets for evaluation and the results are summarized in Table 4. For the regularization-based methods, the translation quality on the incremental language pairs and zero-shot language pairs lags far behind that of the Fine-tuning model. The results indicate that the constraint on the parameters limits the transfer ability of those methods, leading to inferior performance on zero-shot translation. Compared with the PTE model, our method achieves better zero-shot translation performance since we can capture the specific knowledge of the new languages more efficiently.

## 5.3 Sequential Language Adaptation

In this scenario, we enable the original MNMT model to adapt to the incremental language pairs in a sequential manner. We first train the model with the $en{\leftrightarrow}ro$ data and then with the $en{\leftrightarrow}ta$ data. For each method, we extend the embedding layer of the previous model and update the specific hyper-

parameters before training on the new language pairs. For the L2-Reg and EWC methods, we employ the model trained with the $en{\leftrightarrow}ro$ data as the new initial model to compute the regularization loss when continually training the $en{\leftrightarrow}ta$ task. Besides, we also recompute the Fisher information matrix and update the low forgetting risk regions for the EWC and LFR methods, respectively. For our methods, we evaluate the importance of parameters on $en{\leftrightarrow}ta$ and prune the parameters that are important to $en{\leftrightarrow}ta$ task but unimportant to all the previous tasks including $en{\leftrightarrow}ro$ translation. The results are summarized in Table 5. The Fine-Tuning method still encounters the catastrophic forgetting problem so that its performance on $en{\leftrightarrow}ro$ degrades sharply after training with the $en{\leftrightarrow}ta$ data. Although the regularization-based methods achieve relatively better overall performance, they also suffer from performance degradation to different degrees. In contrast to those methods, our method only has a slight decline in the previous tasks and outperforms other continual learning methods on all the translation tasks in this scenario.

## 5.4 Results on Pre-trained MNMT models

We further employ the pre-trained mBART50-nn model (Tang et al., 2020) as the original MNMT model and conduct two incremental learning tasks in the scenario that is closer to the real world.

**Language adaptation task.** In this scenario, we adapt the model to support the Greek↔English (*el↔en*) and Slovak↔English (*sk↔en*). Following Gu et al. (2022), we conduct experiments based on the mBART50-nn+LSE model (Berard, 2021) and report the tokenized BLEU in Table 6. Compared with all the previous continual learning meth-

| Model | Original Language Pairs | | | Incremental Language Pairs | | | | | AVG |
|---|---|---|---|---|---|---|---|---|---|
| | xx→en | en→xx | AVG1 | el→en | en→el | sk→en | en→sk | AVG2 | |
| mBART50-nn+LSE | 26.73 | 21.66 | 24.20 | 27.32 | 16.20 | 36.28 | 28.85 | 27.16 | 25.68 |
| Fine-Tuning | 20.40 | 1.88 | 11.14 | 30.88 | 27.80 | 35.32 | 33.69 | 31.92 | 21.53 |
| L2-Reg | **27.32** | 18.78 | 23.05 | 28.05 | 19.99 | 36.17 | 30.86 | 28.77 | 25.91 |
| EWC | 27.02 | 18.41 | 22.72 | 27.73 | 20.27 | 36.27 | 30.96 | 28.80 | 25.76 |
| LFR | 27.12 | 20.16 | 23.64 | 28.47 | 20.43 | **36.30** | 31.04 | 29.06 | 26.34 |
| Ours(BVP) | 26.58 | **21.58** | **24.08** | **29.62** | **25.64** | 34.80 | **32.29** | **30.59** | **27.34** |

Table 6: The BLEU scores of the language adaptation task. "AVG1" and "AVG2" denote the average BLEU on the original and incremental language pairs, respectively. "AVG" is the overall performance computed by (AVG1+AVG2)/2. We report the tokenized BLEU following Gu et al. (2022). The best scores among all the continual learning methods are highlighted in bold.

| Model | Original Language Pairs | | | Incremental Language Pairs | | | | | AVG |
|---|---|---|---|---|---|---|---|---|---|
| | xx→en | en→xx | AVG1 | de→fr | fr→de | zh→vi | vi→zh | AVG2 | |
| mBART50-nn | 24.88 | 15.63 | 20.26 | 15.88 | 10.17 | 13.35 | 6.48 | 11.47 | 15.87 |
| Fine-Tuning | 0.96 | 2.79 | 1.88 | 29.88 | 28.57 | 30.02 | 30.37 | 29.71 | 15.80 |
| L2-Reg | 20.55 | 15.48 | 18.02 | 29.05 | 23.75 | 23.69 | 25.86 | 25.59 | 21.81 |
| EWC | 22.40 | 15.54 | 18.97 | 29.22 | 23.41 | 23.89 | 26.02 | 25.64 | 22.30 |
| LFR | 22.68 | 15.45 | 19.07 | 30.71 | 24.30 | 24.61 | 26.37 | 26.50 | 22.79 |
| Ours(BVP) | **24.78** | **15.56** | **20.17** | **32.59** | **26.55** | **28.89** | **29.86** | **29.47** | **24.82** |

Table 7: The BLEU scores of the language enhancement task. The best results among all the continual learning methods are highlighted in bold.

ods, our method achieves better overall translation performance on the original and incremental tasks. Specifically, our method narrows the performance gap between the previous continual learning models and the Fine-Tuning model on the incremental language pairs, especially for the en→el direction. These results show that our method can also efficiently adapt to the new language even if it is quite different from the previous languages.

**Language enhancement task.** As mBART50-nn is trained on the English-centric data, the translation quality of non-English language pairs generally lags far behind that of the English-centric language pairs. To this end, we aim to improve the translation performance on two non-English language pairs: German↔French (de↔fr) and Chinese↔Vietnamese (zh↔vi). The results are summarized in Table 7. The regularization-based methods suffer from performance degradation in the xx→en directions, which is different from that in the language adaptation task. We ascribe this to the different training data distributions, where half of the training set uses English as the target during the pre-training stage, while the English sentences

are not available during the incremental learning stage, weakening the ability of the model to translate into English. In contrast to the regularization-based methods, our method still retains the performance on the original language pairs and achieves better translation results on the incremental language pairs.

## 6 Conclusion

In this paper, we propose a dual importance-based model division method in continual learning for MNMT, which is based on the importance of parameters to the original and incremental translation tasks. We search for the parameters that are unimportant to the previous tasks but essential to the incremental tasks and utilize those parameters for new task adaptation. Experimental results show that our method can efficiently adapt the original MNMT model to various incremental translation tasks and consistently outperforms several previous continual learning methods. Further analyses demonstrate that our method can also be applied to the pre-trained MNMT model and benefit the zero-shot translation.

## Limitations

In this work, we attempt to extend an existing MNMT model to support new languages and improve the translation of some language pairs. In addition to the advantages, our method has the following limitations.

(1) Insufficient model capacity. In order to support new translation tasks while keeping the model size fixed, we only utilize a small number of parameters in the original MNMT model for new task adaptation. When adapting to multiple translation tasks simultaneously, the translation performance on the incremental translation tasks still lags behind that of the Fine-Tuning model due to the limited modeling capacity. In the future, we will improve the translation on the new tasks with a better transfer method while not increasing the number of parameters.

(2) Single direction of knowledge transfer. Our method is essentially a parameter isolation-based model in which the parameters contribute differently to different translation tasks. We utilize all the parameters for the translation of new tasks while for the original tasks, we only use the parameters in the pruned model. Therefore, the knowledge can be easily transferred from the original language pairs to the new language pairs. However, the original language pairs can hardly acquire knowledge from the new language pairs. In the future, we will explore methods that can facilitate the knowledge transfer to the original languages to further improve the translation performance on the original tasks.

## Acknowledgements

We sincerely thank all the anonymous reviewers for their insightful comments and suggestions to improve the paper. This work was supported by the National Key Research and Development Program of China (Grant No. 2020AAA0108004) and the Key Research and Development Program of Yunnan Province (Grant No. 202203AA080004).

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

## A Dataset Details

### A.1 WMT-9

**Original Language Pairs.** We select 8 English-centric language pairs of different data sizes from the WMT benchmark to train the original MNMT model that covers 16 translation directions and 9 languages. Specifically, the original language pairs cover languages with different amounts of training data: high-resource (>10M): Russian, Spanish and Japanese; medium-resource (1M~10M): Finnish and Pashto; and low-resource (<1M): Lithuanian, Hindi and Hausa. Moreover, those languages belong to different language branches and vary in several linguistic characteristics such as writing system and dominant word order, which increases the language diversity in the original MNMT model.

**Incremental Language Pairs.** We select another four language pairs for incremental learning considering that the incremental languages are usually non-high resource in the real-world translation scenario. The four languages include: Romanian (low-resource), which is highly related to the original language Spanish with the same language branch Romance, and shares the same script and word order; German (medium-resource), which is similar to the original languages in script and word order, but comes from a different language branch; Tamil (low-resource), which is not related to any original languages and has different scripts; Irish (low-resource), has a completely different word order from all the original languages despite using the shared script of Latin characters.

The characteristics of each language are summarized in Table 8 and the detailed statistics of the training dataset are shown in Table 9.

### A.2 mBART50

We perform incremental learning based on the pre-trained mBART50-nn model for two translation scenarios. The detailed statistics of the training dataset are shown in Table 10.

**Language Adaptation Task.** In this translation scenario, we adapt the model to support the English↔Greek (*en↔el*) and English↔Slovak (*en↔sk*) translation directions following Gu et al. (2022). We use the data from OPUS-100 (Zhang et al., 2020) for model training and the validation/test sets from Flores (Goyal et al., 2022) for checkpoint selection and model evaluation.

| Code | Language | Genus | Script | Order |
|------|----------|-------|--------|-------|
| es | Spanish | Romance | Latin | SVO |
| ru | Russian | Slavic | Cyrillic | SVO |
| ja | Japanese | Japanese | Kanji | SOV |
| fi | Finnish | Finnic | Latin | SVO |
| ps | Pashto | Iranian | Arabic | SOV |
| lt | Lithuanian | Baltic | Latin | SVO |
| hi | Hindi | Indic | Devanagari | SOV |
| ha | Hausa | West Chadic | Latin | SVO |
| ro | Romanian | Romance | Latin | SVO |
| de | German | Germanic | Latin | SVO |
| ta | Tamil | Dravidian | Tamil | SOV |
| ga | Irish | Celtic | Latin | VSO |

Table 8: The characteristics of languages in our experiment. The top half part represents the original languages while the bottom half part represents the incremental languages.

**Language Enhancement Task.** In this translation scenario, we aim to enhance the translation performance of mBART50-nn on German↔French (*de↔fr*) and Chinese↔Vietnamese (*zh↔vi*) directions. The training data are from WMT22[4] and CCMatrix[5] for *de↔fr* and *zh↔vi*, respectively. We also use the validation/test sets from Flores (Goyal et al., 2022) for checkpoint selection and evaluation.

## B Implementation Details

**Vocabulary Extension.** For the experiments on WMT-9, we apply byte pair encoding (BPE) algorithm (Sennrich et al., 2016) using Sentence-Piece (Kudo and Richardson, 2018)[6] to preprocess the sentences of original language pairs with a joint multilingual vocabulary of 64K. For each incremental language, we train a distinct vocabulary of 32K and combine it with the original multilingual dictionary by removing the overlapped tokens. We utilize the combined vocabulary for incremental learning to avoid the out-of-vocabulary problem. As all the methods in our experiments are built on the same extended vocabulary, we do not take these extra parameters into account when comparing the newly added parameters across different methods in this paper.

For the experiments based on mBART50-nn, all the sentences are preprocessed using the Sentence-Piece model provided by XLM-R (Conneau et al., 2020). For the language adaptation task, we insert

---

[4]https://www.statmt.org/wmt22
[5]https://data.statmt.org/cc-matrix
[6]https://github.com/google/sentencepiece

| Language Pair | Data Source | | | # Samples | | |
|---|---|---|---|---|---|---|
| | Train | Valid | Test | Train | Valid | Test |
| en-ru | WMT19 | WMT13 | WMT19 | 38,492,126 | 3000 | 2000 |
| en-es | WMT13 | WMT13 | WMT13 | 15,182,374 | 3003 | 3000 |
| en-ja | WMT21 | WMT20 | WMT21 | 18,012,559 | 993 | 1005 |
| en-fi | WMT19 | WMT18 | WMT19 | 6,587,448 | 3000 | 1996 |
| en-ps | WMT20 | WMT20 | WMT20 | 1,155,944 | 2698 | 2719 |
| en-lt | WMT19 | WMT19 | WMT19 | 635,146 | 3000 | 1000 |
| en-hi | WMT14 | WMT14 | WMT14 | 313,748 | 520 | 2507 |
| en-ha | WMT21 | WMT21 | WMT21 | 752,357 | 2000 | 997 |
| en-ro | WMT16 | WMT16 | WMT16 | 610,320 | 1999 | 1999 |
| en-de | WMT14 | WMT13 | WMT14 | 4,508,785 | 3000 | 3003 |
| en-ta | WMT20 | WMT20 | WMT20 | 660,818 | 1989 | 997 |
| en-ga | OPUS-100 | Flores | Flores | 289,524 | 997 | 1012 |

Table 9: The statistics of train, valid and test data for the original language pairs and incremental language pairs in WMT-9.

| Language Pair | Data Source | | | # Samples | | |
|---|---|---|---|---|---|---|
| | Train | Valid | Test | Train | Valid | Test |
| en-el | OPUS-100 | Flores | Flores | 1M | 997 | 1012 |
| en-sk | OPUS-100 | Flores | Flores | 1M | 997 | 1012 |
| de-fr | WMT22 | Flores | Flores | 18.11M | 997 | 1012 |
| zh-vi | CCMatrix | Flores | Flores | 8.05M | 997 | 1012 |

Table 10: The statistics of train, valid and test data used in continual learning based on mBART50-nn. The top half part represents the incremental language pairs in the language adaptation task while the bottom half part represents the incremental language pairs in the language enhancement task.

a new language-specific embedding layer (LSE) to the original mBART50-nn model to implement the mBART50-nn+LSE model (Berard, 2021) as the baseline model following Gu et al. (2022). For the language enhancement task, we do not need to extend the embedding layer since the languages involved are supported by mBART50-nn.

**Model settings for experiments on WMT-9 dataset.** All the models follow the configuration of Transformer-Big (Vaswani et al., 2017), which consists of 6 stacked encoder/decoder layers and 16 attention heads. The model size $d_{\text{model}}$ and feed-forward dimension $d_{\text{ffn}}$ are set to 1024 and 4096, respectively. For model training, we use the temperature-based sampling strategy to balance the training data distribution with a temperature of $T = 5$ (Arivazhagan et al., 2019), and use *share-all-embeddings* in Fairseq to save parameters. All the model parameters are optimized using Adam optimizer (Kingma and Ba, 2014) ($\beta_1 = 0.9, \beta_2 = 0.98$) with label smoothing of 0.1. The learning rate is scheduled as Vaswani et al. (2017) with a warm-up step of 4000 and a peak learning rate of 0.0005. We train all the MNMT

models on 8 Nvidia RTX A6000 GPUs with a batch of 4096 and adopt the early stop (patience is 10) strategy. For the task-specific fine-tuning in the importance evaluation stage, we fix all the layer normalization layers and the embedding layers in the original MNMT model since we find that it brings better results. For model pruning, we also do not prune the layer normalization layers and embedding layers in the encoder and decoder.

**Model settings for experiments based on mBART50-nn.** The mBART50-nn model consists of 12 stacked encoder/decoder layers and 16 attention heads. The model size $d_{\text{model}}$ and feed-forward dimension $d_{\text{ffn}}$ are set to 1024 and 4096, respectively. Other model configurations are the same as those in the experiments on WMT-9.

**Pruning ratio settings.** Based on the results in Figure 1, in order to retain the performance on the original tasks, the hyper-parameter $a\%$ is set to 25% in all the experiments. For the BVP method, we tune the hyper-parameter $b\%$ in the range of $B = [5\%, 25\%, 50\%, 75\%]$ on each incremental language pair. The final pruning ratios are summarized in Table 12. For the PTE, UMP and UVP

| ID | Method | Embedding ($\theta_E$) | Parameters ($\theta_P$) | $en\leftrightarrow ro$ | $en\leftrightarrow de$ | $en\leftrightarrow ta$ | $en\leftrightarrow ga$ | AVG |
|-----|--------|------------------|-------------------|---------|---------|---------|---------|------|
| (1) | Random-Random | Random | Random | 0.13 | 0.04 | 0.05 | 0.04 | 0.06 |
| (2) | Transfer-Random | Transfer | Random | 0.23 | 0.12 | 0.05 | 0.06 | 0.07 |
| (3) | Random-Transfer | Random | Transfer | 29.16 | 26.27 | 13.16 | 18.48 | 21.76 |
| (4) | Copy-Copy | Copy | Copy | 29.25 | 25.97 | 13.27 | 18.13 | 21.68 |
| (5) | Transfer-Transfer | Transfer | Transfer | 29.29 | 26.04 | 13.31 | 18.36 | 21.75 |

Table 11: The performance on the incremental language pairs under different initialization methods.

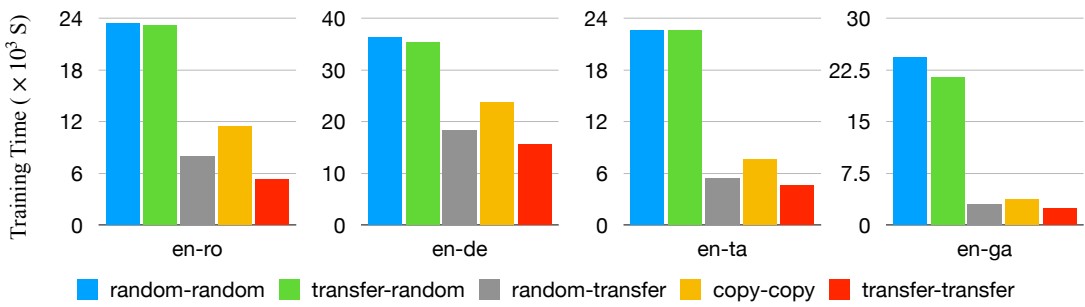

Figure 4: The training time of the BVP model with different parameter initialization methods in incremental learning for MNMT.

| Language Pair | 5% | 25% | 50% | 75% |
|---------------|-----|-----|-----|-----|
| $en\leftrightarrow ro$ | 1.2 | 6.1 | 12.3 | 18.6 |
| $en\leftrightarrow de$ | 1.2 | 6.0 | 12.2 | 18.5 |
| $en\leftrightarrow ta$ | 1.2 | 6.1 | 12.3 | 18.6 |
| $en\leftrightarrow ga$ | 1.3 | 6.4 | 12.6 | 18.8 |

Table 12: The pruning ratios $\rho\%$ of each language pair along with different hyper-parameter $b\%$. The hyper-parameter $a\%$ is set to 25%.

methods, we keep the hyper-parameter $\rho\%$ the same as the pruning ratio of the BVP method for fair comparisons. In our experiments, the hyper-parameter $b\%$ is set to 50% when adapting the original MNMT model to multiple language pairs simultaneously (Table 2 in Section 4.3). Otherwise, $b\%$ is set to 25% in other experiments.

## C  More Results

**Effects of different initialization methods**

As illustrated in Section 3.3, when adapting to the incremental tasks, we add new parameters to the pruned model to expand the model to its original size and extend the embedding layer of the original MNMT model to avoid the out-of-vocabulary problem. In this experiment, we investigate the effects of different initialization methods on the translation performance of the incremental language pairs. Specifically, we study the three initialization meth-

ods:

- **Random**: the newly added parameters or embedding layer are randomly initialized with Gaussian distribution.

- **Copy**: the newly added parameters are initialized with their counterparts in the original MNMT model and the extended embeddings are initialized with the embeddings of the original MNMT model by sampling.

- **Transfer**: the newly added parameters and the extended embeddings are initialized with their counterparts in the fine-tuned task-specific model.

The results are summarized in Table 11. We find that the translation results are more sensitive to the initialization method for the new parameters ($\theta_P$), compared with the embedding layer ($\theta_E$). Specifically, when the new parameters ($\theta_P$) are initialized randomly, it fails to adapt the model to the incremental language pairs. This result suggests that the representation gap between the original parameters and newly added parameters cannot be eliminated during incremental training. By contrast, the copy initialization and transfer initialization methods do not suffer from this problem since the parameters have been well trained to fit the model in advance. The initialization method for the embedding layer

does not apparently affect the translation performance on the incremental language pairs.

We also study the training time of the model with different initialization methods. As shown in Figure 4, the transfer initialization method can remarkably reduce the training time of incremental learning, which is more efficient than other methods. Based on the above results, we adopt the transfer method to initialize the embedding layer ($\theta_P$) and new parameters ($\theta_E$) in the experiments.