# OpenReview forum: "Continual Learning for Multilingual Neural Machine Translation via Dual Importance-based Model Division"
_EMNLP/2023/Conference — EMNLP 2023 Main_

### Official Review · Reviewer_eCJC · 2023-08-05

**Soundness:** 4

**Excitement:**

4: Strong: This paper deepens the understanding of some phenomenon or lowers the barriers to an existing research direction.

**Paper Topic And Main Contributions:**

This paper studies the incremental learning of new languages in multilingual NMT (MNMT). Specifically, to effectively learn new languages while preserving translation performance on original languages, it proposes a dual importance-based parameter division method. The experimental results exhibit competetive BLEU in both new and original languages, across various translation directions and experimental settings (e.g., adding multiple languages or sequentially adding languages).

**Questions For The Authors:**

- Is there any ablation study about dual-importance? (e.g., a comparison with single-step pruning?)
- While the paper provided results about sequential training of two new languages, are there any results with sequential training involving more languages (e.g., 10)? Does BVP still outperform conventional continual learning methods such as EWC?
- In addition to the adaptor, is there any comparison with recently proposed parameter-efficient fine-tuning techniques?
- Is there any reason for freezing embedding and layernorm parameters? Adding detailed explanations would help for a deeper understanding.

**Reasons To Accept:**

- This paper is an interesting work on bridging pruning and incremental learning.
- The effectiveness of the proposed method is supported with thorough experiments and analysis.
- Their method provides promising translation performance compared to previous incremental learning techniques.

**Reasons To Reject:**

- Their method requires complex training pipelines, necessitating an additional training phase for model pruning.

**Reproducibility:**

5: Could easily reproduce the results.

**Reviewer Confidence:**

4: Quite sure. I tried to check the important points carefully. It's unlikely, though conceivable, that I missed something that should affect my ratings.

---

> ### Author Rebuttal · Authors · 2023-08-29
>
> Thanks for your insightful feedback! We address your questions below:
>
> **Q1: Ablation study about dual-importance**
>
> > 1. Is there any ablation study about dual-importance? (e.g., a comparison with single-step pruning?)
>
> We perform the ablation study on the dual-importance pruning strategy as you suggested. The pruning ratio for each method is 6% and the results are shown below. When pruning only based on the old task importance, the model can maintain its performance on the original language pairs. However, it is weak in the ability of capturing the specific knowledge of new language pairs, leading to inferior performance to our BVP model. In contrast, when pruning only based on the new task importance, although the model performs slightly better than BVP on the new language pairs, it fails on the original language pairs. We find that most (>98%) pruned parameters in this method are also important to the old language pairs. Therefore, the performance on original language pairs decreases apparently if those parameters are pruned. Our BVP model achieves the best overall performance across all language pairs, showing the necessity of the dual-importance.
>
>
> | Method                                |  en-xx  | xx-en   |  en-ro  |  en-de  |   en-ta    |   en-ga |
> | :-                                          |     -:      |  -:        |  -:        |  -:          |  -:          |  -:         |
> | Only old task importance    |   18.87  |  23.04  | 28.73 |  25.37  |   13.08   |  19.51 |
> |Only new task importance   |    0.03  |   0.04  |   29.36 |  26.98  |   13.07  |   20.96 |
> Ours (BVP)                          |   18.86  | 23.10   | 29.29    | 26.04  |   13.31  |   20.89 |
>
> **Q2: Sequential training for more languages**
>
> > 2. While the paper provided results about sequential training of two new languages, are there any results with sequential training involving more languages (e.g., 10)? Does BVP still outperform conventional continual learning methods such as EWC?
>
> We further sequentially adapt the model to en-de and en-ga based on the experiment in Table 4 (Section 5.3). Due to the time limitation, we cannot adapt the model to more language pairs. We compare our method with EWC and the results are as follows. As is shown, our BVP model can still balance the performance between original and new language pairs, and consistently outperforms the EWC model.
>
> Results after adapting to en-de:
>
> | Method       |     en-xx   |    en-ro    |    en-ta     |    en-de   |
> | :-                 |     -:          |  -:            |  -:             |  -:           |
> |Scratch         |     21.06   |    27.36   |   12.18     |    27.46  |
> |EWC            |     18.21   |     25.59  |      11.65   |     17.15  |
> |Ours(BVP)   |      20.77   |    28.87   |     12.72   |     25.90   |
>
> Results after adapting to en-ga:
> | Method       |     en-xx   |    en-ro    |    en-ta     |    en-de   |    en-ga  |
> | :-                 |     -:          |  -:            |  -:             |  -:           |  -:           |
> |Scratch       |     21.06    |    27.36   |     12.18   |     27.46  |     12.52  |
> |EWC           |     17.72    |    24.65   |    11.13    |     16.22  |     11.11   |
> |Ours(BVP)  |     20.67   |     28.83  |     12.69   |      25.27 |      11.87   |
>
> **Q3: More comparisons with other parameter-efficient fine-tuning method**
>
> > 3. In addition to the adaptor, is there any comparison with recently proposed parameter-efficient fine-tuning techniques?
>
> We evaluate some recent parameter-efficient fine-tuning (PEFT) methods including prompt-tuning [1], prefix-tuning [2] and LoRA [3] on en-ro language pair. The results are as follows. Similar to the Adapter model, those PEFT methods do not degrade the performance on original language pairs since they employ additional parameters for en-ro adaptation. However, they yield inferior performance to the Adapter model on the new language pairs (en-ro) and also underperform our BVP model.
>
> |Method      |   en$\rightarrow$xx    |     xx$\rightarrow$en   |    en$\rightarrow$ro   |    ro$\rightarrow$en  |
> | :-                |     -:          |  -:            |  -:             |  -:           |
> |Prompt       |  18.90    |    23.21     |       19.85      |    30.20   |
> |Prefix          |   18.90   |     23.21      |      20.24     |     30.48   |
> |LoRA          |   18.90   |     23.21      |      22.95      |    31.86   |
> |Adapter       |   18.90   |    23.21     |        23.89      |    33.36   |
> |Ours(BVP)  |    18.86   |     23.10     |       24.60     |     33.98   |
>
> **Q4: Explanations for freezing embedding and layernorm parameters**
>
> > 4. Is there any reason for freezing embedding and layernorm parameters? Adding detailed explanations would help for a deeper understanding.
>
> We freeze the embedding and layernorm parameters during the training stage since we find that it benefits the task-specific finetuning in the parameter importance evaluation stage and yields better final translation performance on new language pairs. Moreover, Gu et al. [4] find that it helps to alleviate the forgetting problem on the original language pairs. We will make it clear in the updated version as you suggested.
>
>
> [1] Lester et al. The power of scale for parameter-efficient prompt tuning. In EMNLP2021.
>
> [2] Li and Liang. Prefix-tuning: Optimizing continuous prompts for generation. In ACL2021.
>
> [3] Hu et al. LoRA: Low-rank adaptation of large language models.
>
> [4] Gu et al. Continual Learning of Neural Machine Translation within Low Forgetting Risk Regions. In EMNLP2022.

---

### Official Review · Reviewer_5Yj3 · 2023-08-06

**Soundness:** 3

**Excitement:**

2: Mediocre: This paper makes marginal contributions (vs non-contemporaneous work), so I would rather not see it in the conference.

**Paper Topic And Main Contributions:**

This paper presents an algorithm to address the issue of catastrophic forgetting in language translation tasks. The proposed approach involves a three-step process that balances the performance of the model on both original and new language pairs.

The first step involves fine-tuning a pre-trained model to identify parameters that are essential for new tasks but have minimal impact on the original tasks. This results in a pruned model. In the second step, the pruned model is expanded with additional parameters, which are then fine-tuned using new training data. Experimental results demonstrate that this approach is effective in retaining the performance of the model on original tasks while also improving its performance on new translation tasks.

**Questions For The Authors:**

see above.

**Reasons To Accept:**

This paper introduces an algorithm designed to address the problem of catastrophic forgetting in language translation tasks.

The algorithm is straightforward and easy to implement, making it a practical solution for mitigating this issue. By following the steps outlined in the proposed approach, it is possible to improve the performance of new tasks while minimizing the impact of catastrophic forgetting.



**Reasons To Reject:**

1. **The experimental results of fine-tuning presented in this paper are not convincing.** For instance, in Table 1, the performance of the Original Language Pairs after fine-tuning decreased from 23.21 to 0.50. Similar phenomena can also be observed in Tables 2, 5, and 6. A BLEU score of 0.5 indicates that the model no longer has knowledge of this language. Such scores are difficult to believe.  **Furthermore, fine-tuning is an important step in identifying important parameters.**


2. **Further analysis is needed to determine the importance of various parameters.** The calculation of importance is based on the absolute change value, which raises several concerns:
+ The range of change for different parameters is related to their location. For example, the change in high-level parameters may be greater than that of embedding parameters.
+ Selecting parameters based on their absolute value may lead to over-adjustment. In an extreme case where no adjustment is needed, selecting parameters based on their absolute value may still result in some being adjusted. How can we determine which parameters are truly important for the new task and have minimal impact on the original model?

3. This paper tries to extend an existing MNMT model to support new languages and improve the translation of some language pairs.  There are many pre-trained MNMT models available, such as M2M-100, NLLB, and BLOOM. However, this paper trains a model from scratch or uses mBART50-nn (2020). **The rationale for this selection is unclear.**


**Reproducibility:**

4: Could mostly reproduce the results, but there may be some variation because of sample variance or minor variations in their interpretation of the protocol or method.

**Reviewer Confidence:**

3: Pretty sure, but there's a chance I missed something. Although I have a good feel for this area in general, I did not carefully check the paper's details, e.g., the math, experimental design, or novelty.

---

> ### Author Rebuttal · Authors · 2023-08-29
>
> Thanks for your insightful comments. We address each of your concerns below:
>
> **Regarding the results of Fine-Tuning method**
>
> > 1. The experimental results of fine-tuning presented in this paper are not convincing. For instance, in Table 1, the performance of the Original Language Pairs after fine-tuning decreased from 23.21 to 0.50. Similar phenomena can also be observed in Tables 2, 5, and 6. A BLEU score of 0.5 indicates that the model no longer has knowledge of this language. Such scores are difficult to believe. Furthermore, fine-tuning is an important step in identifying important parameters.
>
> We would like to clarify the translation results of the Fine-Tuning method. As described in line 949 in Appendix B, **the Fine-Tuning method trains the incremental language pairs based on the pre-trained MNMT model only with the training data of new language pairs (the training data of original language pairs are not available)**. In this setting, the model will only focus on learning the knowledge of new language pairs while forgetting the knowledge of original language pairs, which is known as the catastrophic forgetting problem. Therefore, the model can hardly output correct target words or fluent sentences when translating across original language pairs, leading to the remarkable performance decline (in Tables 1, 2, 5, 6). Such serious performance decline is also reported in some recent studies (Gu et al. [1], Huang et al. [2] and Zhao et al. [3]). Take the results in Table 5 for example, our results on original language pairs are similar to those reported by Gu et al. [1] under the same experimental settings, and we put the comparisons blow. Moreover, Huang et al. [2] also reported that the performance on original language pairs drops to 1.14 BLEU from 15.14 BLEU for en-xx direction (-14.00 BLEU) and 1.68 BLEU from 20.19 BLEU for xx-en direction (-18.51 BLEU). Given the above comparisons, the results of our Fine-Tuning method are reasonable. The Fine-tuning method is actually an important step in identifying important parameters for new language pairs and it will not be affected by the poor performance on the original language pairs.
>
> |Method                   |    xx$\rightarrow$en    |    en$\rightarrow$xx    |
> |:-                             |      -:                            |                -:                   |
> |mBART50-nn+LSE (Gu et al. [1])  |   25.83   |  21.48 |
> |Fine-Tuning (Gu et al. [1])     |    18.37  |   1.15  |
> |mBART50-nn+LSE (Ours reimplemetation)      |   26.73      |   21.66   |
> |Fine-Tuning  (Our reimplemetation)       |     20.40    |     1.88   |
>
>
> **Further analysis on the importance of the parameters**
>
> > 2.1. The range of change for different parameters is related to their location. For example, the change in high-level parameters may be greater than that of embedding parameters.
>
> We study the parameter change value in the en-ro adaptation scenario to show more analysis about our method. We fine-tune the original MNMT model on en-ro training data and calculate the average value change of each weight matrix. As you say, the range of change for different parameters is actually related to their location and the change in high-level parameters tends to be greater than that of the low-level parameters. However, the changes for all the weight matrices are mostly in the range of (0.015, 0.025). Similar results can also be found in language adaptation task (e.g. en-ta, en-de). These results show that all the weight matrices contribute to the new language pair, so we consider all the weight matrices for weight pruning. Despite that the changes in some weight matrices are relatively larger, we could not prune all the parameters in those weight matrices since those parameters may also be important to the original language pairs. Otherwise, the performance on original language pairs will decline remarkably. To this end, we only prune the parameters that are important to the new tasks while not important to the old tasks to make a compromise between the old and new tasks. Moreover, we can increase the hyper-parameter $b$ in our BVP method to add more parameters to the pruning candidate $\mathcal{N}$ to cover more parameters with larger value changes. In fact, we find that the final pruning ratios of each weight matrix are different from each other in our BVP method, which also indicates that the effects of various parameters are related to their location.
>
> > 2.2. Selecting parameters based on their absolute value may lead to over-adjustment. In an extreme case where no adjustment is needed, selecting parameters based on their absolute value may still result in some being adjusted. How can we determine which parameters are truly important for the new task and have minimal impact on the original model?
>
> As the analysis on en-ro shows, all the weight matrices contribute to the new language pairs. Therefore, the adjustment in each weight matrix is necessary. We agree that some parameters may not need to be adjusted since the performance on new language pairs will not be further improved when the pruning ratio is larger than 12% (See Figure 3(b)). We try to alleviate this problem by carefully tuning the pruning ratio. Previous studies (Gu et al. [4] and Xie et al. [5]) also try other methods such as Taylor Expansion for model pruning, however, the performance is not always better than the absolute value method.
>
> **Regarding the pre-trained model selection**
>
> > 3. This paper tries to extend an existing MNMT model to support new languages and improve the translation of some language pairs. There are many pre-trained MNMT models available, such as M2M-100, NLLB, and BLOOM. However, this paper trains a model from scratch or uses mBART50-nn (2020). The rationale for this selection is unclear.
>
> We start our experiments from a self-trained MNMT model since we can select the languages during the pre-training and incremental learning stages, which makes it easier for us to study some characteristics that are possible to affect the incremental learning results in MNMT such as scripts, dominant order, data size and language relatedness. Our results show that the translation performance on incremental language pairs is actually related to those characteristics. We select mBART50-nn as the pre-trained model to perform incremental learning in the real-world scenario. Adapting the original model to new languages is regarded as one important task for the incremental learning in MNMT. Although the newly released pre-trained MNMT models such as M2M100 and NLLB are also available, it is difficult to further add new languages to them because they have covered over 100 languages. Moreover, our experiments on mBART50-nn can also be implemented in other pre-trained models such as M2M100. Given the above, we implement our method based on the mBART50-nn and perform two incremental tasks (language adaptation task and language enhancement task) that are the most common cases in the real world.
> Despite that, we perform incremental learning of de-fr and zh-vi language pairs based on the M2M100 (1.2B) model as you suggested. The results are as follows. We only report the performance on en-xx language pair for original tasks due to the time limitation. Our BVP model still consistently outperforms the vanilla M2M100 model on new language pairs without causing obvious performance decline on the original language pairs. The results show that our method generalizes to other pre-trained models.
>
> |Method    |    xx-en  |	en-xx   |    fr-de    |   de-fr   |   zh-vi	   |    vi-zh    |
> |  :-           |    -:         |      -:          |     -:         |    -:       |     -:        |      -:       |
> |M2M100(1.2B)	|  23.54 |	16.99    |   28.19 |    35.42   |   28.42 |	27.66 |
> |Ours(BVP)           |   23.47  |	16.96 |	29.36  |   35.94   |   29.68 |	30.40 |
>
>
> [1] Gu et al. Continual Learning of Neural Machine Translation within Low Forgetting Risk Regions. In EMNLP2022.
>
> [2] Huang et al. Knowledge Transfer in Incremental Learning for Multilingual Neural Machine Translation. In ACL2023.
>
> [3] Zhao et al. Life-long Learning for Multilingual Neural Machine Translation with Knowledge Distillation
>
> [4] Gu et al. Pruning-then-Expanding Model for Domain Adaptation of Neural Machine Translation. In NAACL2021.
>
> [5] Xie et al. Importance-based Neuron Allocation for Multilingual Neural Machine Translation. In ACL2021.

---

### Official Review · Reviewer_Tywa · 2023-08-10

**Soundness:** 3

**Excitement:**

4: Strong: This paper deepens the understanding of some phenomenon or lowers the barriers to an existing research direction.

**Missing References:**

- Missing related works utilizing dynamic adaptation of MNMT models:
 -Garcia et al. Towards continual learning for multilingual machine translation via vocabulary substitution
 -Lakew et al., Transfer learning in multilingual neural machine translation with dynamic vocabulary

**Paper Topic And Main Contributions:**

This work propose an multilingual NMT model fine-tuning and parameter pruning approaches, with the goal to improve new language pair (LP) translation while maintaining quality for pre-existing LP's. Different from other works, authors propose model parameter weighting or importance evaluation strategies to prune parameters. Three different pruning approaches are discussed by evaluating parameters importance for pre-existing LP's and new LP. Experimental results show, the proposed approach to improve translation for new LP's.

**Questions For The Authors:**

- In some instances (Table 1) the performance on original LP's degrade with significant bleu scores (>1). Can the authors rephrase some statements or clearly state the cases where the proposed approach degrade for pre-existing LP's?

**Reasons To Accept:**

- Incremental model tuning strategy doesn't require original training data, which could have been a constraint for leveraging most pre-trained MNMT models.
- Paper is well written, authors did a great job making the work  easily understandable, along with the illustration provided.
- Extensive experiment and discussions.

**Reasons To Reject:**

- Model expansion step seems to defeat the original motivation of the work (i.e. not adding additional parameters when tuning a model to new translation direction)
- Missing experiments for LP's unseen in the parent model, this has been regarded as one of the mai challenge previous works have been trying to address. See comments for the rest.

**Reproducibility:**

3: Could reproduce the results with some difficulty. The settings of parameters are underspecified or subjectively determined; the training/evaluation data are not widely available.

**Reviewer Confidence:**

4: Quite sure. I tried to check the important points carefully. It's unlikely, though conceivable, that I missed something that should affect my ratings.

**Typos Grammar Style And Presentation Improvements:**

- L103: "Finally, we utilize expand the pruned model ..."? remove utilize ?

---

> ### Author Rebuttal · Authors · 2023-08-29
>
> Thanks for your insightful comments! We address your concerns below:
>
> **Regarding additional parameters**
>
> > 1. Model expansion step seems to defeat the original motivation of the work (i.e. not adding additional parameters when tuning a model to new translation direction)
>
> In the model expansion step, we introduce two newly added parameters: new parameter $\theta_P$ and new embedding $\theta_E$. The new parameter $\theta_P$ is used to expand the pruned model to its original size so that model parameters are not increased. The new embedding $\theta_E$ is used only when the incremental languages are not supported by the original MNMT model. In this case, it increases the number of model parameters in the embedding layer. In our experiments, we add the new embedding $\theta_E$ for all the baselines by default on WMT-9 dataset to avoid the out-of-vocabulary problem. Therefore, we do not take the embeddings into account when discussing the additional parameters. We will make it clear in the updated version to avoid misunderstanding.
>
> **Regarding the missing experiments for unseen language pairs**
>
> > 2. Missing experiments for LP's unseen in the parent model, this has been regarded as one of the mai challenge previous works have been trying to address.
>
> We discuss three translation scenarios of unseen language pairs in our experiments. First, adapting the parent model to new languages. In this scenario, we adapt the WMT-9 model to four new languages (ro, de, ta, ga) and perform translation between those four languages and English (en) (See Table 1). Second, based on the first scenario, we further perform translation from non-English languages in the parent model to the new languages (See Table 3). In this scenario, the language pairs are also unseen in the parent model (e.g. es-ro). Third, adding language pairs that the parent model is not trained on their supervised data (See Table 6). For example, de and fr are already supported by mBART50-nn, but the mBART50-nn model has not been trained on de-fr data and it performs translation between de and fr in zero-shot fashion. Therefore, the de-fr language pair also belongs to unseen language pairs in the parent model.
>
> **Regarding the questions**
>
> > 1. In some instances (Table 1) the performance on original LP's degrade with significant bleu scores (>1). Can the authors rephrase some statements or clearly state the cases where the proposed approach degrade for pre-existing LP's?
>
> We would like to explain why the performance on original language pairs degrades with a significant BLEU in Table 1. When adapting the original MNMT model to new language pairs, the L2-Reg, EWC and LFR methods fine-tune all the parameters in the model. Although they utilize regularizer or hard constraint to let the parameters stay close to their original values, the change of all parameters still results in the performance decline to some extent, especially when the new languages are similar to the original languages. For example, when adapting to en-ro language pair, the original language pair en$\rightarrow$es declines over 10 BLEU which is remarkably larger than other language pairs, leading to over 1 average BLEU decline on the en-xx direction in the original translation task. For our method, we prune a small group of parameters in the original MNMT model based on the parameter importance. Similar to previous studies, the pruning step will naturally result in some performance decline. However, the performance loss for each original language pair is relatively smaller (at most 0.3 BLEU).
>
> **Missing references**
>
> Thanks for pointing out these related studies. We will include a discussion of these studies in our updated version based on your feedback.
>
> **Typos**
>
> Thanks for catching the typos! The word “utilize” in L103 should be removed. We have fixed it in the revision.

---

### Official Review · Reviewer_5LWe · 2023-08-11

**Typos Grammar Style And Presentation Improvements:** none
**Soundness:** 3

**Excitement:**

3: Ambivalent: It has merits (e.g., it reports state-of-the-art results, the idea is nice), but there are key weaknesses (e.g., it describes incremental work), and it can significantly benefit from another round of revision. However, I won't object to accepting it if my co-reviewers champion it.

**Missing References:**

none

**Paper Topic And Main Contributions:**

The research problem in this paper is the alleviate the catastrophic forgetting problem during learning the new translation tasks for new languages for the MNMT model. To do that, the authors propose a method named dual importance-based model division to divide the model parameters for old tasks and new tasks respectively. Experiment results demonstrate the effectiveness of the proposed method.

**Questions For The Authors:**

A. How do you balance the parameters between the old tasks and new tasks, if they share very similar features? Such as the transition tasks for close languages. In this situation, the weights of the fine-tuning model on the new tasks may share highly similar to the original tasks.

**Reasons To Accept:**

1. The paper is easy to follow.

2. The proposed method the catastrophic forgetting problem to some extent as indicated by the experiment results, through freezing the original parameters.

**Reasons To Reject:**

1. The original tasks’ performance degenerates to some extent and underperforms the baseline of the Adapter, which indicates the negative influence of removing some parts of the original networks.

2. The proposed method may encounter a limitation if the users continuously add new languages because of the limited model capacity.

**Reproducibility:**

3: Could reproduce the results with some difficulty. The settings of parameters are underspecified or subjectively determined; the training/evaluation data are not widely available.

**Reviewer Confidence:**

3: Pretty sure, but there's a chance I missed something. Although I have a good feel for this area in general, I did not carefully check the paper's details, e.g., the math, experimental design, or novelty.

---

> ### Author Rebuttal · Authors · 2023-08-29
>
> Thanks for your insightful comments! We address your concerns below:
>
> **Regarding the degradation on original language pairs**
> > 1. The original tasks’ performance degenerates to some extent and underperforms the baseline of the Adapter, which indicates the negative influence of removing some parts of the original networks.
>
> Our method does encounter a slight performance decline compared with the Adapter method, because our method utilizes a small group of parameters in the original MNMT model to learn the knowledge of the new language pairs, while the Adapter method fixes the whole original MNMT model. However, unlike the Adapter mothed, our method does not require extra parameters for incremental learning. Moreover, the performance decline is much smaller than that of the previous replay-based and regularization-based methods, which is relatively acceptable. The performance decline can also be alleviated by reducing the pruning ratio or increasing the model capacity, as shown in Figure 1.
>
> **Regarding the limited model capacity**
>
> >2. The proposed method may encounter a limitation if the users continuously add new languages because of the limited model capacity.
>
> The capacity bottleneck is a common problem in MNMT when many languages are accommodated in a single model. In our case, there are two possible solutions to alleviate that problem. One solution is to directly enlarge the model capacity. As shown in Figure 1, although mBART50-nn covers much more languages than our WMT-9 model (50 vs. 9), it suffers from smaller performance degradation since it has a larger model capacity. The results show that there are more unimportant parameters in the mBART50-nn model that can be used to adapt to more new language pairs. The other solution is to vary the pruning ratios for different incremental language pairs. As shown in Figure 3(b), the BLEU score on incremental language pairs does not increase when the pruning ratio is larger than 12%. Therefore, we can allocate different pruning ratios to different language pairs in order to accommodate more new language pairs.
>
> **Regarding the questions**
>
> > A. How do you balance the parameters between the old tasks and new tasks, if they share very similar features? Such as the transition tasks for close languages. In this situation, the weights of the fine-tuning model on the new tasks may share highly similar to the original tasks.
>
> For the cases where the new languages are very similar to the original languages, we can utilize a smaller pruning ratio to retrain more common features between original and new languages. Although similar languages share more common features, Lin et al. [1] show that they still differ from each other on their language specific sub-networks. In our experiment on WMT-9 dataset, Spanish (es) is one of the original languages and we adapt the original MNMT model to Romanian (ro) and German (de), respectively. The three languages are similar as they are all from the Indo-European language family. The translation performance on en-es is not apparently affected when adapting to en-ro and en-de (The performance on en-es declines <0.1 BLEU). Moreover, we find that the overlap in the pruned parameters of en-ro and en-de only accounts for 26.8%, which means that 73.2% of the pruned parameters are non-overlapping, indicating there is a big difference between en-ro and en-de language pairs. Our fine-tuning model can capture those difference and yield different pruned parameters. Given the above observations, our method can deal with the case where the old tasks and new tasks share similar features.
>
> [1] Lin et al. Learning Language Specific Sub-network for Multilingual Machine Translation. In ACL2021.

---

### Meta-Review · Area_Chair_MeGN · 2023-09-14

**Recommendation:** 4

**Metareview:**

This paper addresses the problem of adding language pairs to multilingual MT models. It proposes
a method that detectes parameters in the base model which are both unneeded for the original
languages, and suitable for fine-tuning in the incremental languages. Experiments show that the
method achieves bleu improvements on the incremental languages, whilst mostly preserving
performance on the original languages.

The reviewers are in broad agreement about the soundness of the paper, with two 3s and two 4s. I cannot
see any substantive issues raised by the reviewers which would justify the 3 score.
he issue that I would raise is that
this paper's claims and conclusions depend heavily on bleu scores. I would expect to see additional
results on a trainable neural metric (eg comet) to validate the results.

There is some disagreement about the excitement, with reviewer 5Yj3 giving the paper a 2. However their
first 2 reasons to reject seem to be based on misunderstandings, and the third reason (the choice
of base model) is addressed in the rebuttal. Both eCJC and Tywa rate the excitement at 4 and liked
the interesting method, and the thorough experiments and analysis.

Overall, an interesting paper, with strong experiments, except for their over-reliance on bleu scores.

---

### Decision · Program_Chairs · 2023-10-07

**Decision:**

Accept-Main

**Comment:**

This paper addresses the problem of adding language pairs to multilingual MT models. It proposes
a method that detectes parameters in the base model which are both unneeded for the original
languages, and suitable for fine-tuning in the incremental languages. Experiments show that the
method achieves bleu improvements on the incremental languages, whilst mostly preserving
performance on the original languages.

The reviewers are in broad agreement about the soundness of the paper, with two 3s and two 4s. I cannot
see any substantive issues raised by the reviewers which would justify the 3 score.
he issue that I would raise is that
this paper's claims and conclusions depend heavily on bleu scores. I would expect to see additional
results on a trainable neural metric (eg comet) to validate the results.

There is some disagreement about the excitement, with reviewer 5Yj3 giving the paper a 2. However their
first 2 reasons to reject seem to be based on misunderstandings, and the third reason (the choice
of base model) is addressed in the rebuttal. Both eCJC and Tywa rate the excitement at 4 and liked
the interesting method, and the thorough experiments and analysis.

Overall, an interesting paper, with strong experiments, except for their over-reliance on bleu scores.